# Human Health Risk Assessment for Toxic Trace Elements in the Yaro Mine and Reclamation Options

**DOI:** 10.3390/ijerph16245077

**Published:** 2019-12-12

**Authors:** Min-Suk Kim, Mi Jeong Park, Jeong Hwa Yang, Sang-Hwan Lee

**Affiliations:** 1O-Jeong Eco Resilience Institute, Korea University, Seoul 02841, Korea; adoniss8686@gmail.com; 2Technical Research Institute, Mine Reclamation Corporation, Wonju 26464, Korea; pmj3382@mireco.or.kr (M.J.P.); jhyang@mireco.or.kr (J.H.Y.)

**Keywords:** mine, risk assessment, reclamation, soil, toxic trace elements

## Abstract

The aim of this study was to investigate the environmental impact and human health risks associated with toxic trace element (TTE) exposure in the abandoned Yaro Mine, Korea. Carcinogenic and non-carcinogenic risks were assessed separately for adults and children. Among the various pathways, the rate of TTE intake from the ingestion of groundwater was highest, followed in descending order by crop consumption, soil ingestion, and soil contact. The carcinogenic risk from the ingestion of groundwater was highest, followed by crop consumption and ingestion of contaminated surface soil. The non-carcinogenic risk from the ingestion of groundwater was highest (53.57% of the total non-carcinogenic risk), followed by crop intake (38.53%) and surface soil ingestion (4.80%). The risk assessment revealed that contaminated soil around Yaro mine posed a high risk to the health of inhabitants, mainly via groundwater ingestion and crop consumption. Reclamation measures should include methods of disrupting the high-risk routes between the source and recipient. Stabilization and covering techniques are promising options for reducing the hazard (i.e., exposure to the bioavailable fraction of TTE) and creating a chemical or physicochemical barrier to the potential migration pathways.

## 1. Introduction

Mining is one of the major anthropogenic sources of contamination in the environment. In areas where these activities are concentrated, various toxic trace elements (TTE), including arsenic (As) and other toxic heavy metals, could be released to the surrounding environment through acid mine drainage from mine tailing [1,2]. TTE released from mining sites could increase the concentration of aquatic and soil environments, which can have a detrimental effect on biota and human health [3,4].

In Korea, the Ministry of Trade, Industry, and Energy (KMoTIE) informed that approximately 5400 mines existed. More than 85% of them are left unmanaged without any minimum safety precautions, resulting in the contamination of environment and local inhabitants [5]. Especially, mining and refining facilities at abandoned mine areas have been left unmanaged, with mine tailings and waste rocks scattered randomly. Therefore, the management of abandoned derelict mines has become a major concern. Korean society has learned from past experiences that there is an urgent need to mitigate the risks imposed by the mining of soils to ensure sustainability of the environment and protect inhabitant’s health [6].

In Korea, several reclamation practices have been widely applied to contaminated soil abandoned mine areas since 2006. Followed by domestic legislative in Korea, both types of sites require remediation and/or reclamation: (1) those where soil contamination exceeds the ‘‘trigger value’’ and (2) those where soil contamination exceeds the Korean legislative limit for edible crops. Whereas lead exposure to humans is diverse in other countries, such as electronic waste, ceramics, and traditional medicines [7], the mine is the main exposure route in Korea.

The use of risk assessment techniques in mining areas has focused primarily on human health issues rather than soil reclamation [5,8,9,10]. In the case of arsenic and lead, which are also listed as a domestic limit for edible crops, several studies have been conducted on the effects on the human body, such as the cardiovascular system, renal system, hepatic system, etc. [11,12,13]. In addition, when children are exposed to lead, they continue to have negative effects not only in childhood but also in adulthood [14]. However, a few studies have proposed remediation and/or reclamation methods based on risk assessment results.

The aims of this study were to (1) determine TTE contamination levels in upland soil contaminated by mining activity, (2) assess the potential health risks to farmers and children associated with exposure to TTE of concern, and (3) propose reclamation options to reduce the risk.

## 2. Materials and Methods

### 2.1. Study Area

The Yaro gold mine is located in Hapcheon Province in Gyungnamdo, Korea (Figure 1). The climate of this region is humid and hot in summer and cold and dry in winter, with an annual average temperature of 13.4 °C and rainfall of 1340 mm. The Yaro mine was developed to extract gold in the 1930s. After the mine closed in 1994, mine tailings and waste rocks have been left without any adequate environmental management. Mining wastes in this area were proliferated into the surrounding cultivated land by water and wind action.

### 2.2. Sample Collection

A total of 37 surface soil (0–30 cm) samples were collected. At each sampling site, five subsamples were collected and homogenized for a composite sample, which was immediately transported to the laboratory. In this area, groundwater is used as drinking water, and therefore groundwater samples were collected from six wells. Rice, red pepper, and garlic were also sampled because they were produced in relatively large amounts in the study area.

### 2.3. Sample Analysis

Pseudo-total TTE concentrations, including arsenic (As), cadmium (Cd), copper (Cu), lead (Pb), and zinc (Zn) were determined by wet-digestion procedure with aqua regia, (a mixture of HCl/HNO_3_ (*v:v* = 3:1), followed by ISO 11466 [15], and the bioavailable fraction of TTEs was determined following NH_4_NO_3_ extraction according to ISO 19730 [16]. All the TTE in solution were analyzed using and inductively coupled plasma optical emission spectrometer (ICP-OES, Agilent, Santa Clara, CA, USA).

Groundwater samples were filtered through a 0.45 μm membrane and then acidified with concentrated HNO_3_. The TTE concentrations in the water samples were measured using inductively coupled plasma mass spectrometry (ICP-MS). Crop samples were collected from farmlands and paddy fields around the mine. Harvested plants were rinsed with distilled water and dried at 80 °C for 48 h. The plants were digested with HNO_3_ + H_2_O_2_ using a hot-block digestion procedure at 120 °C. The resultant solution was diluted with 1% (*v/v*) nitric acid. The trace elements concentrations in the edible parts of crop plants were determined by ICP-OES.

Certified reference materials (NIST 2711a and NIST 1568b) were included in the analysis to ensure internal quality assurance/quality control. The recovery rates and relative standard deviation were measured. The recovery in the reference soil and crops was between 80% and 120%, and the relative standard deviation was less than 20%.

### 2.4. Risk Assessment

The potential adverse effects of the Yaro mine area on human health through the exposure to TTE were calculated using the multiphase and multicomponent risk assessment model that was developed for the Korean soil contamination risk assessment guidelines [17].

The following exposure routes were considered: (1) soil (by direct ingestion of soil particles or dermal adsorption), (2) water (by direct ingestion of groundwater), and (3) crops (by consumption of contaminated crops). There are several exposure models available that describe the intake of TTE through above pathways [18,19]. Most of the models are found on similar assumptions and equations and the equations and required parameters used in this study are showed in Table 1 and Table 2.

A well-known route of exposure is the direct intake of dirt or dust, especially for children who have a habit of often putting hands and fingers in the mouths. Van Wijnen et al. [22] reported that children ingest approximately 150–200 mg soil per day up to the age of 6 years. Although no specific studies are conducted in adults, published guidelines for risk assessment studies suggest that a soil limit of 50 to 100 mg per day is recommended [19]. In this study, the ingestion rate of soil for children and adults was 118 and 50 mgday^−1^, respectively [23]. Dust inhalation means the inhalation of particles smaller than 10 μm. In general, large particles are swallowed and contribute directly to the route of ingestion. Important parameters are the amount of inhaled air per day, the concentration of suspended particles in the air, and the percentage of remained particles in the lungs [24]. Exposure through skin contact means the absorption of contaminants into the bloodstream after contact with the skin. The calculation was based on the chemical specific factors of the amount of attached soil per surface area of the skin and the absorption of TTE through the skin, which is exposed daily [19]. In this study, the surface areas for adults and children were 4212 and 2978 cm^2^, respectively [25]. Intake of crops means the consumption of crops cultivated in contaminated soils. An important parameter used was the concentration of TTE in the edible parts of the crop plant. The consumption rates of rice, pepper, and garlic for adults were 0.161, 0.005, and 0.001 kg day^−1^, respectively [23].

Carcinogenic risk has been estimated to increase the likelihood that a person will develop cancer for life as a result of exposure to potential carcinogens. The following linear low dose carcinogenic risk equations were used for each exposure route [17,18]:
Carcinogenic risk = ADI × SF,(1)
where ADI is the average daily intake averaged over 70 years (mg kg^−1^ day^−1^) and SF is the slope factor. If a site contained multiple carcinogenic contaminants, the carcinogenic risks for each carcinogen and each exposure route were combined (i.e., assuming an additivity of effects) and compared with the acceptable risk. Risks in the range of 10^−6^ to 10^−4^ are typically considered acceptable by the KMoE [17].

The HQ (hazard quotient) is the ratio between the ADI of TTE and the reference dose for a given contaminant (RfD):(2)$$\mathrm{HQ}\;=\frac{\mathrm{ADI}}{\mathrm{RfD}},$$
where ADI is the estimated ADI for a given contaminant (mg kg^−1^ day^−1^) and RfD (mg kg^−1^ day^−1^) is the reference dose for daily exposure to a particular TTE via a route that does not lead to significant detrimental health effects during a person′s lifetime [17]. In order to assess the overall likelihood of non-carcinogenic effects by various chemicals, the calculated HQ values for each chemical are combined (i.e., assuming an additivity of effects), represented by a risk index (HI) [17]:HI = ∑ HQk,(3)
where HI is the sum of all HQ for all exposure routes, HQk is the HQ for exposure route k. If HI is ≤1, the exposed population is considered to have no risk of toxicity from the TTE; if HI is >1, exposure to the TTE may have adverse health effects on human health [26].

## 3. Results and Discussion

### 3.1. Toxic Trace Elements in Soil, Water, and Crops

The concentrations of target TTE in soil, groundwater, and crops are shown in Table 3. All TTE concentrations in the surface soil samples were significantly greater in the study area than in the reference area. The mean concentrations of As, Cd, Cu, Pb, and Zn in soils near the abandoned mines were 31.77, 3.31, 19.87, 93.88, and 288.47 mg kg^−1^, respectively. These results suggested that the elevated TTE concentrations in soils were related to the mining activities, and hence the land may not be suitable for agricultural uses. The concentrations of As, Cd, Cu, Pb, and Zn were 5.09, 23.68, 1.05, 4.68, and 4.06 times greater than the natural levels in agricultural soils in Korea, respectively [27]. The high TTE concentrations measured in the groundwater were As, Cd, Cu, Pb, and Zn higher than the reference values for these chemicals, and therefore represent a health risk when drinking water. The TTE concentrations in agricultural products, e.g., rice, red pepper, and garlic, are presented in Table 3. High TTE concentrations in agricultural products cultivated in contaminated mining areas have also been reported in other studies [4,28]. All of the TTE concentrations were significantly higher in crops in the Yaro mining area than in the reference areas reported by Kim et al. [29]. For example, the background Cd concentration in rice was informed to be 0.019 mg kg^−1^, whereas the rice collected in the Yaro mining area exceeded the maximum permissible level of 0.2 mg kg^−1^ [30], indicated that rice paddy soils near the abandoned mine area might not be suitable for rice cultivation. Cd contamination can lead to health concerns when it occurs in soil and foods, especially rice [31].

### 3.2. Human Health Risk Assessment

Table 4 shows the estimates of average daily metal intake for farmers and children in adults in various media. TTE’s ADI was highest in Zn, followed by Cu, Pb, As, and Cd in descending order. Through various routes, the daily intake of TTE through crop consumption was highest, followed by groundwater ingestion, soil ingestion, soil contact, and soil inhalation. The intake via crop ingestion was highest in rice, followed by the next red pepper and garlic.

Table 5 showed the calculated carcinogenic risks for TTE from various media. The oral slope factors, i.e., ingestion of soil, groundwater, and crops, for As and Pb were 1.50 and 8.50 × 10^−3^, respectively, according to Korean guidelines (Table 6). For As, the slope factor for soil contact (dermal) was 3.3. In case of soil inhalation, the slope factors for As, Cd, and Pb were 4.3 × 10^−3^, 1.80 × 10^−3^, and 2.20 × 10^−5^, respectively. The carcinogenic risk from metals was highest for As (2.51 × 10^−4^, 98.83% of the total carcinogenic risk), followed by Pb (2.93 × 10^−6^) and Cd (4.35 × 10^−8^). The carcinogenic risk according to exposure pathway was highest for groundwater ingestion (64.85% of the total carcinogenic risk), followed in descending order by crop consumption, surface soil ingestion, soil contact, and soil inhalation. Carcinogenic risk is the probability that a person’s lifelong exposure to carcinogenic chemicals causes an individual to develop all types of cancer. Acceptable or tolerable total risk for legal regulation is defined as 10^−5^ in Canada [32] and Spain [33] and a range of 10^−6^ to 10^−4^ in Korea [17]. Considering 10^−5^ as an acceptable risk, i.e., 1/100,000 people, the study area posed a considerable carcinogenic risk due to As contamination. The sum of all carcinogenic risks for adults was 2.54 × 10^−4^. This value was equivalent to a probability of about 2.5 cancers in 10,000 people. The total carcinogenic risk for children was 4.37 × 10^−5^, corresponding to 17.24% of the carcinogenic risk for adults. Followed by Park and Choi [4], the sum of carcinogenic risk for As, Cd, and Pb including ingestion, contact and inhalation of soil, water ingestion and crop ingestion pathways was calculated as 5.81 × 10^−03^, which is 23 times higher than that of this study. And they also calculated the sum of carcinogenic risk for other mining area, with a value of 6.29 × 10^−03^ [34], which is 25 times higher than this study. The reason for the difference is that the risk value for inhalation was calculated very largely using farmers’ coefficients. Taken together, it is a very important step to accurately identify a significant exposure pathway at the site of the target area before conducting a risk assessment.

Table 7 showed the estimated non-carcinogenic risks for TTE from various media. The non-carcinogenic risk (i.e., the HI) for TTE was highest for Pb (0.68), followed by As (0.56), Cd (0.24), Zn (0.11), and Cu (0.06). Among the exposure pathway, the highest non-carcinogenic risk was imposed by ingestion of groundwater (53.57% of the total non-carcinogenic risk), followed in descending order by crop consumption, ingestion of surface soil, contact with soil, and inhalation of soil particles. The HI value of each TTE in each route was less than 1, but the sum exceeded 1 (1.64 for adults). Therefore, the non-carcinogenic risk from TTE in this area was considerable for adults. In the case of children, the HI value of each TTE in each route was less than 1, and their sum was also less than 1 (Table 7). These results suggested that the non-carcinogenic risk from TTE in this area was not substantial for children.

### 3.3. Reclamation Options

According to this human health risk assessment, the Yaro mine soils has been found to threaten a significant risk to the health of inhabitants, primarily through direct ingestion of groundwater and consumption of crops. Reclamation measures should include methods to reduce the high-risk routes between the source and recipient, or actions to remove the source of the hazard from the abandoned area. Among the various types of techniques, in Korea, one of the major reclamation techniques is removing the contaminated top soil and replacing it with clean soil [35]. Therefore, the contaminated soil is updated with a layer of clean soil that is thick enough to limit exposure to TTE via hand-to-mouth or dermal contact. In addition, chemical stabilization methods reducing the mobility and availability of TTE could be another option using various types of amendments [36,37]. Compared to other reclamation methods, in situ chemical stabilization is less expensive and might provide a long-term remediation solution through the adsorption, formation of low-solubility minerals, or precipitates [36]. Therefore, it is suitable for the reclamation of broad areas of agricultural soil or low-value land. The proposed soil reclamation method involved an incorporation of these two strategies, i.e., management of mobility and/or availability of contaminants in soil using amendments and covering the contaminated soil with clean soil (Figure 2). The stabilization effect (i.e., reducing the bioavailability of TTE) was confirmed in this study. Addition of 3% (*w*/*w*) of limestone reduced NH_4_NO_3_-extractable As, Cd, Cu, Pb, and Zn by 16%, 66%, 44%, 96%, and 66%, respectively. In situ stabilization, including the conversion of the bioavailable TTE species to less soluble species through the precipitation or ion-exchange mechanisms, was conducted using limestone, widely used representative amendment. In addition to limestone, acid mine drainage sludge, red mud, furnace slag, etc. are also actively examined and proposed for chemical stabilization [37,38]. These would result in a significant reduction in the release of contaminants to the aquatic or biotic environment.

## 4. Conclusions

The present study was conducted to confirm the environmental impact and human health risks associated with TTE in the abandoned Yaro Mine, Korea. For the risk assessments, carcinogenic and non-carcinogenic risks were assessed separately for adults and children. The rate of trace elements intake from the ingestion of groundwater was highest, followed in descending order by crop consumption, soil ingestion, and soil contact. The carcinogenic risk from the ingestion of groundwater was highest, followed by crop consumption and ingestion of contaminated surface soil. The non-carcinogenic risk from the ingestion of groundwater was highest (53.57% of the total non-carcinogenic risk), followed by crop intake (38.53%) and surface soil ingestion (4.80%). These results indicated that the risk assessment revealed that contaminated soil around Yaro mine posed a high risk to the health of inhabitants, mainly via groundwater ingestion and crop consumption. For the successful gentle remediation of agricultural abandoned mining area, among the various restoration methods, stabilization and covering techniques are promising options for decreasing the hazard and creating a chemical and/or physicochemical barrier to the potential migration pathways through the food chain.

## Figures and Tables

**Figure 1 ijerph-16-05077-f001:**
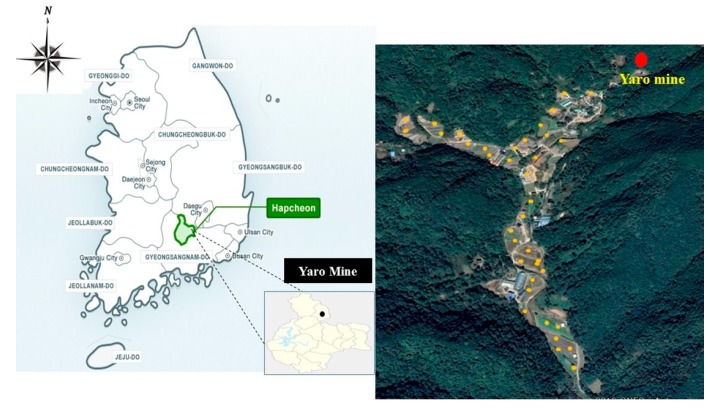
Location of the Yaro mine and sampling points.

**Figure 2 ijerph-16-05077-f002:**
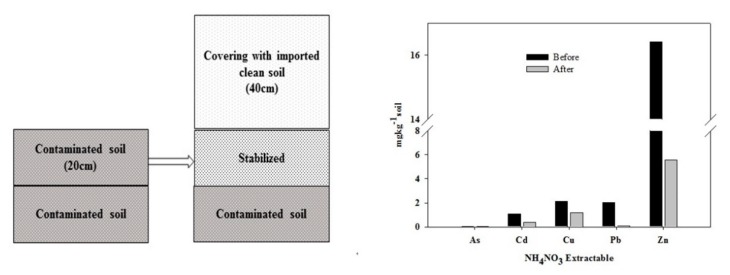
A schematic of a soil reclamation method (**Left**) and the reduced bioavailable fraction caused by soil stabilization (**Right**).

**Table 1 ijerph-16-05077-t001:** Equations used to determine the average daily intake for different exposure pathways.

Exposure Pathway	Equation for Average Daily Intake ^a^
Surface soil oral ingestion (mg kg^−1^ day^−1^)	(C _s_ × CR _s_ × EF × ED)/(BW × AT)
Surface soil dermal (mg kg^−1^ day^−1^)	(C_s_ × AF × SA_e_ × EF × ED)/(BW × AT)
Inhalation of surface soil (mg m^−3^)	(C_s_ × TSP × frs × F_r_ × EF × ED)/(BW × AT)
Intake of groundwater (mg kg^−1^ day^−1^)	(C_w_ × CR_w_ × EF × ED)/(BW × AT)
Intake of crop (mg kg^−1^ day^−1^)	(C_p_ × CR_p_ × EF × ED)/(BW × AT)

^a^ C = concentrations of toxic trace elements in each medium, other parameters and factor are listed in Table 2; C_s_, concentration of contaminants in soil; CR_s_, intake rate of soil; EF, exposure frequency; ED, exposure duration; BW, body weight; AT, average time; AF, soil-skim adsorption coefficient; SA_e_, surface area body for dermal contact of soil; TSP, amount of air-suspended particles; Fr, retention of soil particles in lung

**Table 2 ijerph-16-05077-t002:** Exposure parameters and factors used for adult farmers and children.

Factors/Parameters		Unit	Acceptor		References
			Adults	Children	
Body weight (BW)		kg	59.9	36	KNIER, 2011
Exposure duration (ED)		years	48.4	8.2	KNIER, 2011
Average time (AT)		days	28,656	29,908	KOSIS, 2013 [20]
Exposure frequency (EF)		days year^−1^	350	350	KMoE, 2009
Surface area body for dermal contact of soil (SA_e_)		cm^2^ day^−1^	4212	2978	KNIER, 2011
Soil-skin adsorption coefficient (AF)		mg cm^−2^	0.07	0.2	USEPA, 2009 [21]
Amount of air-suspended particles (TSP)		mg m^−3^	0.07	0.07	KNIER, 2011
Soil fraction in particles in the air (frs)		-	0.5	0.5	KNIER, 2011
Retention of soil particles in lung (Fr)		-	0.75	0.75	KMoE, 2009
Intake rate of soil (CR_s_)		mg day^−1^	50	118	KMoE, 2007
Intake rate of groundwater (CR_w_)		L day^−1^	1.66	1.00	KMoE, 2007
Intake rate of crop (CR_c_)	Rice	kg day^−1^	0.161	0.108	KMoE, 2007
	Red pepper		0.005	0.004	
	Garlic		0.001	0.001	

**Table 3 ijerph-16-05077-t003:** Toxic trace elements (TTE) concentrations in environmental media around the abandoned Yaro metal mine.

Type		No. of Samples	Units	Trace Elements				
				As	Cd	Cu	Pb	Zn
Soil ^a^		37	mg kg^−1^	31.77	3.31	19.87	93.88	288.47
Water		6	mg L^−1^	0.01	0.01	0.01	0.01	0.10
Crops	Rice	6	mg kg^−1^	0.03	0.40	7.77	0.09	19.45
	Red pepper	7	mg kg^−1^	0.04	0.08	1.23	ND ^b^	8.95
	Garlic	7	mg kg^−1^	0.25	0.17	2.00	ND	8.56

^a^ Background level of As, Cd, Cu, Pb, and Zn in agricultural soil in Korea is 6.24, 0.14, 18.86, 20.07, and 71.07 mg kg^−1^ soil, respectively, ^b^ Not detected.

**Table 4 ijerph-16-05077-t004:** Average daily intake of toxic trace elements (TTE) from each medium.

Life Stage	Pathway	As	Cd	Cu	Pb	Zn
		mg kg^−1^ day^−1^				
Adults	Surface soil oral ingestion	2.72 × 10^−06^	1.92 × 10^−08^	5.36 × 10^−06^	3.46 × 10^−05^	9.12 × 10^−05^
	Surface soil contact	5.06 × 10^−07^	6.35 × 10^−07^	5.54 × 10^−06^	1.22 × 10^−06^	1.08 × 10^−05^
	Surface soil inhalation	9.00 × 10^−08^	2.42 × 10^−08^	2.96 × 10^−07^	1.09 × 10^−06^	2.87 × 10^−06^
	Groundwater oral ingestion	1.09 × 10^−04^	7.79 × 10^−05^	1.56 × 10^−03^	1.71 × 10^−04^	1.60 × 10^−03^
	Crop oral ingestion	5.43 × 10^−05^	1.53 × 10^−05^	6.73 × 10^−03^	1.36 × 10^−04^	3.01 × 10^−02^
	Total	1.67 × 10^−04^	9.39 × 10^−05^	8.30 × 10^−03^	3.44 × 10^−04^	3.18 × 10^−02^
Children	Surface soil oral ingestion	1.73 × 10^−06^	1.23 × 10^−08^	3.42 × 10^−06^	2.21 × 10^−05^	5.82 × 10^−05^
	Surface soil contact	2.76 × 10^−07^	3.47 × 10^−07^	3.03 × 10^−06^	6.69 × 10^−07^	5.88 × 10^−06^
	Surface soil inhalation	1.46 × 10^−08^	3.93 × 10^−09^	4.80 × 10^−08^	1.77 × 10^−07^	4.66 × 10^−07^
	Groundwater oral ingestion	1.77 × 10^−05^	1.27 × 10^−05^	2.54 × 10^−04^	2.79 × 10^−05^	2.61 × 10^−04^
	Crop oral ingestion	8.63 × 10^−06^	8.63 × 10^−06^	1.22 × 10^−03^	2.61 × 10^−04^	5.37 × 10^−03^
	Total	2.84 × 10^−05^	2.17 × 10^−05^	1.48 × 10^−03^	3.12 × 10^−04^	5.70 × 10^−03^

**Table 5 ijerph-16-05077-t005:** Carcinogenic risks for toxic trace elements (TTE) in the Yaro mine area.

Life Stage	Pathway	As	Cd	Cu	Pb	Zn	Total
Adults	Surface soil oral	4.08 × 10^−06^	-	-	2.94 × 10^−07^	-	4.37 × 10^−06^
	Surface soil dermal	1.67 × 10^−06^	-	-	-	-	1.67 × 10^−06^
	Surface soil inhalation	3.87 × 10^−07^	4.35 × 10^−08^	-	1.31 × 10^−08^	-	4.44 × 10^−07^
	Groundwater oral	1.63 × 10^−04^	-	-	1.46 × 10^−06^	-	1.64 × 10^−04^
	Crop oral	8.15 × 10^−05^	-	-	1.16 × 10^−06^	-	8.27 × 10^−05^
	Total	2.51 × 10^−04^	4.35 × 10^−08^	-	2.93 × 10^−06^	-	2.54 × 10^−04^
Children	Surface soil oral	2.60 × 10^−06^	-	-	1.88 × 10^−07^	-	2.79 × 10^−06^
	Surface soil dermal	9.12 × 10^−07^	-	-		-	9.12 × 10^−07^
	Surface soil inhalation	6.28 × 10^−08^	7.07 × 10^−09^	-	2.12 × 10^−09^	-	7.20 × 10^−08^
	Groundwater oral	2.66 × 10^−05^	-	-	2.37 × 10^−07^	-	2.68 × 10^−05^
	Crop oral	1.29 × 10^−05^	-	-	2.09 × 10^−07^	-	1.31 × 10^−05^
	Total	4.31 × 10^−05^	7.07 × 10^−09^	-	6.36 × 10^−07^	-	4.37 × 10^−05^

**Table 6 ijerph-16-05077-t006:** The slope factors (mg kg^−1^ day^−1^)^−1^ and reference doses (mg kg^−1^ day^−1^) applied in this study.

Classification	Slope Factor	As	Cd	Cu	Pb	Zn
Carcinogenic	Oral slope factor (SF_o_) (mgkg^−1^-day)^−1^	1.50	ND ^a^	ND	8.50 × 10^−03^	ND
	Dermal slope factor (SF_abs_) (mgkg^−1^-day)^−1^	3.30	ND	ND	ND	ND
	Inhalation unit risk (URF_inh_) (㎍m^−3^)^−1^	4.30 × 10^−03^	1.80 × 10^−03^	ND	1.20 × 10^−05^	ND
Non-carcinogenic	Oral reference dose (Rf_o_) (mgkg^−1^-day)	3.00 × 10^−04^	5.00 × 10^−04^	1.40 × 10^−01^	5.00 × 10^−04^	3.00 × 10^−01^
	Dermal reference dose (RfDabs) (mgkg^−1^-day)^−1^	2.90 × 10^−04^	1.30 × 10^−05^	ND	ND	ND
	Inhalation reference dose (RfC) (mgm^−3^)	ND	7.00 × 10^−04^	1.00 × 10^−03^	ND	ND

^a^ Not detected.

**Table 7 ijerph-16-05077-t007:** Non-carcinogenic risks for toxic trace elements (TTE) in the Yaro mine area.

Life Stage	Pathway	As	Cd	Cu	Pb	Zn	Total
Adults	Surface soil oral	9.06 × 10^−03^	3.85 × 10^−05^	3.83 × 10^−05^	6.92 × 10^−02^	3.04 × 10^−04^	7.86 × 10^−02^
	Surface soil dermal	1.75 × 10^−03^	4.88 × 10^−02^	-	-	-	5.06 × 10^−02^
	Surface soil inhalation	-	3.46 × 10^−05^	2.96 × 10^−04^	-	-	3.31 × 10^−04^
	Groundwater oral	3.63 × 10^−01^	1.56 × 10^−01^	1.11 × 10^−02^	3.43 × 10^−01^	5.35 × 10^−03^	8.78 × 10^−01^
	Crop oral	1.81 × 10^−01^	3.06 × 10^−02^	4.81 × 10^−02^	2.72 × 10^−01^	1.00 × 10^−01^	6.32 × 10^−01^
Children	Surface soil oral	5.78 × 10^−03^	2.45 × 10^−05^	2.44 × 10^−05^	4.42 × 10^−02^	1.94 × 10^−04^	5.02 × 10^−02^
	Surface soil dermal	9.53 × 10^−04^	2.67 × 10^−02^	-	-	-	2.77 × 10^−02^
	Surface soil inhalation	-	5.61 × 10^−06^	4.80 × 10^−05^	-	-	5.36 × 10^−05^
	Groundwater oral	5.92 × 10^−02^	2.54 × 10^−02^	1.81 × 10^−03^	5.58 × 10^−02^	8.71 × 10^−04^	1.43 × 10^−01^
	Crop oral	2.88 × 10^−02^	5.51 × 10^−03^	8.69 × 10^−03^	4.93 × 10^−02^	-	9.23 × 10^−02^

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
