# Peer review of "Human Health Risk Assessment for Toxic Trace Elements in the Yaro Mine and Reclamation Options"

_ijerph, 2019, doi:10.3390/ijerph16245077_

Round 1
Reviewer 1 Report
The paper is centered in the risk assessment for adults and children exposed to an abandoned mine “Yaro mine”in Korea, via soil inhalation, ingestion, groundwater intake and crop ingestion, contaminated with As, Cd, Cu, Pb and Zn taking in account carcinogenic and non-carcinogenic effects. In general, the manuscript is well written and the results and discussion are clear. The manuscript is suitable for its publication after minor changes.
Major corrections
In table 5 and 7, they are showing a column and a row labeled “total” as a result of adding all the individual risks results in horizontal (adding all the results independent of the element) and in vertical (adding all the results for a particular element). I understand their meaning in table 5, since you can assess carcinogenic risk taking in account all the routes of exposure for an element or all the probabilities independently of the element since the only effect is cancer. However, in table 7 the column labeled “total” has no sense, since you cannot asses the sum of the individual risk of the all the elements, due that the non-carcinogenic effects are particular for each element in a particular concentration. I suggest to remove this column unless you have a good explanation for use it in the manuscript.
Minor corrections
Line 151-154
Replace “Gallic” for “Garclic” at the bottom of the table
Author Response
First of all, thank you very much for your sincere review of my manuscript.
Table 5 was edited followed by reviewer's comment.
And misspelling was also corrected.
Thank you very much.
Sincerely.

Reviewer 2 Report
The introduction is too short and simply not as good as it can be.
Please frame the introduction by stating how bad metals are for human health and how the soil carries the legacy of their exposure due to numerous sources of lead accumulating in the soil from gasoline, paint, and other sources.
Cite articles such as:
Obeng-Gyasi, E., 2019. Sources of lead exposure in various countries. Reviews on environmental health, 34(1), pp.25-34.
Then, talk about some of the health outcomes that can come about from exposure to these metals citing articles such as:
1) cardiovascular system
Lanphear, Bruce P., Stephen Rauch, Peggy Auinger, Ryan W. Allen, and Richard W. Hornung. "Low-level lead exposure and mortality in US adults: a population-based cohort study." The Lancet Public Health 3, no. 4 (2018): e177-e184.
Obeng-Gyasi, E., 2019. Lead Exposure and Cardiovascular Disease among Young and Middle-Aged Adults. Medical Sciences, 7(11), p.103.
2) Renal system
Harari, Florencia, Gerd Sallsten, Anders Christensson, Marinka Petkovic, Bo Hedblad, Niklas Forsgard, Olle Melander et al. "Blood Lead Levels and Decreased Kidney Function in a Population-Based Cohort." American Journal of Kidney Diseases (2018).
Lin, Ja-Liang, Dan-Tzu Lin-Tan, Kuang-Hung Hsu, and Chun-Chen Yu. "Environmental lead exposure and progression of chronic renal diseases in patients without diabetes." New England Journal of Medicine 348, no. 4 (2003): 277-286.
3) Hepatic system
Obeng-Gyasi, Emmanuel, Rodrigo X. Armijos, M. Margaret Weigel, Gabriel Filippelli, and M. Aaron Sayegh. "Hepatobiliary-Related Outcomes in US Adults Exposed to Lead." Environments 5, no. 4 (2018): 46.
Can, S., C. BaÄŸci, M. Ozaslan, A. I. Bozkurt, B. Cengiz, E. A. Cakmak, R. KocabaÅŸ, E. KaradaÄŸ, and M. TarakçioÄŸlu. "Occupational lead exposure effect on liver functions and biochemical parameters." Acta Physiologica Hungarica 95, no. 4 (2008): 395-403.
also speak to why we care about exposure to these metals which is the fact that they can affect individuals for their life-course.
4) Reuben, A., Caspi, A., Belsky, D.W., Broadbent, J., Harrington, H., Sugden, K., Houts, R.M., Ramrakha, S., Poulton, R. and Moffitt, T.E., 2017. Association of childhood blood lead levels with cognitive function and socioeconomic status at age 38 years and with IQ change and socioeconomic mobility between childhood and adulthood. Jama, 317(12), pp.1244-1251.
5) Obeng-Gyasi, E., 2018. Lead exposure and oxidative stress—A life course approach in US adults. Toxics, 6(3), p.42.
After that, you can start on your topic. This will make reading the article more enjoyable and pique readers interest.
Methods:
Line 88- dermal absorption
Overall, the methods are really well done.
Results and Discussion:
This section needs more comparison between the results of this study and other similar studies. That will give readers a better gauge of the implications of the findings.
Overall this work is very critical and with a few changes it will be ready for publication.
Author Response
First of all, thank you very much for your sincere review of my manuscript.
In manuscript, several references were added followed by reviewer's comment and recommendation to reinforce the purpose and necessity of research.
And misspelling was also edited.
Thank you very much.
Sincerely.

Round 2
Reviewer 2 Report
Well done on corrections. The paper is much improved.